# Investigation of wear behaviour and surface analysis of a coated H13 material for friction drilling application

Saravanan Balakrishnan[1], Selvakumar Subbaiah[2]*, Mathew Alphonse[3], Robert Čep[4], Sachin Salunkhe[5,6], Emad Abouel Nasr[7]

**1** Department of Mechanical Engineering, Sri Ramakrishna College of Engineering, Perambalur, Tamil Nadu, India, **2** Department of Mechanical Engineering, M.A.M School of Engineering, Siruganur, Tiruchirappalli, Tamil Nadu, India, **3** Department of Mechanical Engineering, Vel Tech Rangarajan Dr. Sagunthala R&D Institute of Science and Technology, Chennai, Tamil Nadu, India, **4** Department of Machining, Assembly and Engineering Metrology, Faculty of Mechanical Engineering, VSB-Technical University of Ostrava, Ostrava, Czech Republic, **5** Department of Biosciences, Saveetha School of Engineering, Saveetha Institute of Medical and Technical Sciences, Chennai, India, **6** Department of Mechanical Engineering, Gazi University Faculty of Engineering, Ankara, Turkey, **7** Department of Industrial Engineering, College of Engineering, King Saud University, Saudi Arabia

* sdy.12051972@gmail.com

## Abstract

In recent years, industries have seen many advancements in finding proper tools for machining to enhance productivity. Choosing a proper friction drilling tool that minimizes surface damage and improves tool life and productivity is essential. In this study, the wear characteristics of H13 steel among four samples (untreated, heated, TiAlN, and AlCrN) were investigated through a pin-on-disc machine, focusing on highlighting the wear behaviour and surface morphology. The novelty of this study is to analyze an optimal friction drilling tool that can enhance its life. The tempering process was carried out to improve the hardness of the H13 steel tool from 37 HRC to 57 HRC. During the wear test process, the temperature is maintained at 250°C. Using an Atomic Force Microscope (AFM), the worn surface of the samples was analyzed. Among the four samples (untreated, heated, TiAlN, and AlCrN), the untreated samples were affected by adhesive wear and oxidation. It is observed that the tempering helps the coated H13 samples to appear wear-resistant; the material loss obtained for the coated samples is much less compared to the uncoated samples. The untreated and heated sample CoF values observed are 0.713 and 0.591; for TiAlN and AlCrN, the CoF values observed are 0.481 and 0.416. This study reveals that AlCrN Coated H13 steel exhibited the best wear response. Hence, it is suitable for Friction drilling applications.

**Data availability statement:** All relevant data are within the manuscript.

**Funding:** The authors present their appreciation to King Saud University for funding this research through the Ongoing Research Funding program (ORF-2025-164), King Saud University, Riyadh, Saudi Arabia. This article was co-funded by the European Union under the REFRESH – Research Excellence For Region Sustainability and High-tech Industries project number CZ.10.03.01/00/22_003/0000048 via the Operational Programme Just Transition and has been done in connection with project Students Grant Competition SP2024/087 Specific Research of Sustainable Manufacturing Technologies "financed by the Ministry of Education, Youth and Sports and Faculty of Mechanical Engineering VŠB-TUO. The article has been done in connection with the project Students Grant Competition SP2024/087", Specific Research of Sustainable Manufacturing Technologies "financed by the Ministry of Education, Youth and Sports and Faculty of Mechanical Engineering VŠB-TUO.

**Competing interests:** The authors have declared that no competing interests exist.

## 1. Introduction

Coating on materials is crucial in improving performance and durability [1,2]. Industries are finding a suitable abrasive, oxidation, and adhesion wear solution. In the case of adhesion wear, the defect occurs due to the surface bonding; due to this, a large amount of material transfers and delamination happens [3,4]. Because of the more rigid and softer material mating, the surface quality might go high in abrasive wear [5,6]. In oxidation, the brittle oxides are formed based on oxygen reaction. Given that a better solution is needed, an effective strategy like selecting the appropriate material and surface treatment is essential to reduce these types of wear [7–9]. It is observed that recent trends in nanotechnology support in ensuring against corrosion and wear. Moreover, the coating industries look forward to promoting sustainability by depositing eco-friendly coating [10,11]. The coating helps extend material life, prevents thermal damage, and is heat resistant. Few coatings like titanium nitride have confirmed that they can reduce friction, resulting in better surface finish material [12–15]. Another advantage of coating is reducing the usage of coolants, which helps promote an environmental practice [16,17]. Telasang et al. studied the magnetron sputtering method; they reported that coatings like AlCr and AlCrFe are intricately structured, offering various advantages like corrosion resistance and wear [18]. Chayeuski et al. reported that coatings like titanium nitride (TiN) and diamond-like carbon coating can improve the wear resistance and hardness of the material. In this investigation, the bonding between the coating and substrate was analyzed, and the strong bonding between the surfaces improved the tools' life [19,20]. Mishra et al. have examined WC/Co material usage on surfaces. The samples were coated using AlCrN and AlTiN, respectively. The findings were compared with untreated steel, indicating a 30% improvement in wear behaviour [21]. In another study, Arrabal et al. found that the alloys could be better protected using Plasma electrolytic oxidation (PEO) coating; the wear rate was lower, up to 30%. The coating results have proved an increase in hardness and porosity reduction. Further, the friction coefficient has improved concerning the rise in load [22]. In another study, the content of TaC and TiC was increased and added to substrates, which led to an improvement in the wear behaviour of a TiAlN-coated tool. Moreover, the results showed a 20% reduction in wear rate and a 15% increase in tool hardness [23]. Due to the deposition of TiAlN in the substrate, it is observed that the adhesion strength and hardness improved, rising from 16.7 N and 24.6 GPA to 17.3 N and 30.1 GPA, respectively [24,25]. It is also essential to study parent materials, especially the research and usage and characteristics of H13 steel, including toughness and wear resistance, which make H13 useful in the industry and extend the tool's life.

Furthermore, Due to its excellent hot hardness, wear resistance, and toughness, H13 tool steel is the best choice for friction drilling. The H13 steel can maintain hardness at elevated temperatures, minimizing tool wear and deformation, which is critical for consistent bush and bosh formation. Adopting coating will help reduce the risk of surface damage and withstand higher temperatures [26–28]. In addition, the coating helps to serve as a barrier against abrasion, thermal fatigue, and friction [29–31]. Friction drilling is an innovative method in the hole-making process; the production

industry faces many challenges in saving the tool life, particularly in the heat-generated process. Coating plays a significant role in reducing heat generation; it helps migrate heat and friction. Also, coating enhances the tool's hardness, which could help improve the tool's wear resistance. The process parameters like spindle speed and load play a crucial role in machining, directly affecting the coefficient of friction (CoF). The advantage of the friction drilling process is the formation of bosh and bush with the help of the tool, while during the tool penetration into the work material due to the heat generated, the material softens and extrudes the top and bottom of the workpiece can be called as extruded collar and elongated sleeve.

This study aims to focus on choosing a better coating for preparing friction drilling tools. Based on its low cost, the H13 steel was chosen for the investigation. It is essential to analyse the wear behaviour. The tribological analysis was compared with H13 steel when TiAlN and AlCrN were deposited.

## 2. Experimental procedure

The steel tool's chemical composition is stated in Table 1. Initially, the hardness is about 28 HRC; after the quenching and hardening of H13 steel improves the material hardness [32–34]. In the hardening process, the H13 samples were preheated at 950°C to 1050°C, followed by the tempering process between 500–600°C for toughness. This helps refine the steel structure; the hardness has been elevated up to 53–55 HRC, increasing the wear and toughness. The surface heat treatment process typically supports and improves the steel's hardness [35,36].

After the hardening process, the surface of H13 samples was grinded with the help of waterproof abrasive paper to obtain a smooth surface, followed by polishing the surface to minimize surface defects. The deposition was done using the Physical Vapor Deposition (PVD) method [37]. The samples of AlCrN were placed in the vacuum chamber using the magnetron sputtering process. The aluminium 50% and chromium 50% were vaporized using nitrogen gas. In this method, rich nitrogen was allowed to react with aluminium and chromium atoms to form a very hard, thermally stable AlCrN coating on the surface of the H13 sample. During the process, the deposition was controlled to form a uniform thickness. The temperature and pressure were maintained at 550°C and 0.01 bar, respectively. Similarly, the samples of TiAlN were placed in the vacuum chamber using a magnetron sputtering process. The titanium 50% and aluminium 50% were vaporized using nitrogen gas. In this method, rich nitrogen was allowed to react with titanium and aluminium atoms to form a very hard, thermally stable TiAlN coating on the surface of the H13 sample. During the process, the deposition was controlled to form a uniform thickness. The temperature and pressure were maintained at 550°C and 0.01 bar, respectively. The coating thickness observed during deposition was 3.1 μm for TiAlN and 3.55 μm for AlCrN samples.

The H13 steel tool is prepared as a pin (12 mm), and EN31(30 mm) is prepared as disc material. Initially, the disk is mounted on a platform, where the platform rotates at a higher rpm, and the pin is loaded against the disc. The rated parameters for the process are sliding distance, diameter, and velocity of 200 m, 30 mm, and 1.5 m/s, respectively. The pin is loaded up to 20 N based on the above parameters. Speed and time for wear study are calculated as 955 rpm and 10 minutes, respectively. The wear rate and frictional forces were continuously monitored and recorded during the Pin-on-Disc test, as shown in Fig 1. This research has focused on adhesive wear and abrasive wear mechanisms.

After the wear study, scanning electron microscopy (SEM) begins testing the samples [38,39]. SEM and EDS analysis interprets the samples to evaluate the surface morphology, identify defects like cracks, and determine material composition. This study uses energy-dispersive X-ray Spectroscopy (EDS) to analyze the crucial elemental composition with high accuracy of materials [40]. EDS enables quantitative and qualitative identification of elements present in coated

**Table 1. Chemical composition of H13 steel.**

| C | Si | Mn | Cr | Mo | V | Fe |
|---|---|---|---|---|---|---|
| 0.39 | 1.14 | 0.41 | 5.0 | 1.4 | 1.1 | Balance |

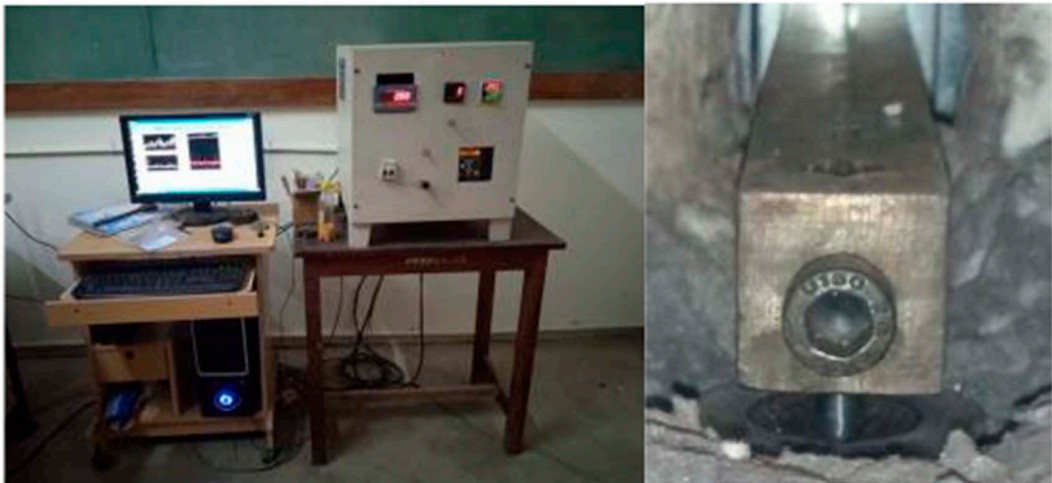

**Fig 1. Pin on disk setup (H13 sample mounted).**

samples. Finally, the powerful tool, Atomic Force Microscope (AFM), is used to find the surface wear characteristics on the nanoscale. A sharp probe on a flexible cantilever scans all samples separately. AFM captures surface material properties and topography with sub-nanometer resolution, providing deep insights into surface roughness. Friction drilling is a five-step process; in the initial step, the rotating conical tool approaches the work material, as shown in Fig 2. In the next stage, due to the high rotation of the tool, friction is generated between the tool and the workpiece. This helps the workpiece to soften and allows the tool to penetrate the workpiece without removing the excess material. As a result, excess material forms at the top and bottom of the workpiece, known as bosh and bush, which supports holding and acts as a washer. In the final stage, the tool is retracted, which helps improve the surface finish. The whole process helps minimize the waste, and no chips are formed.

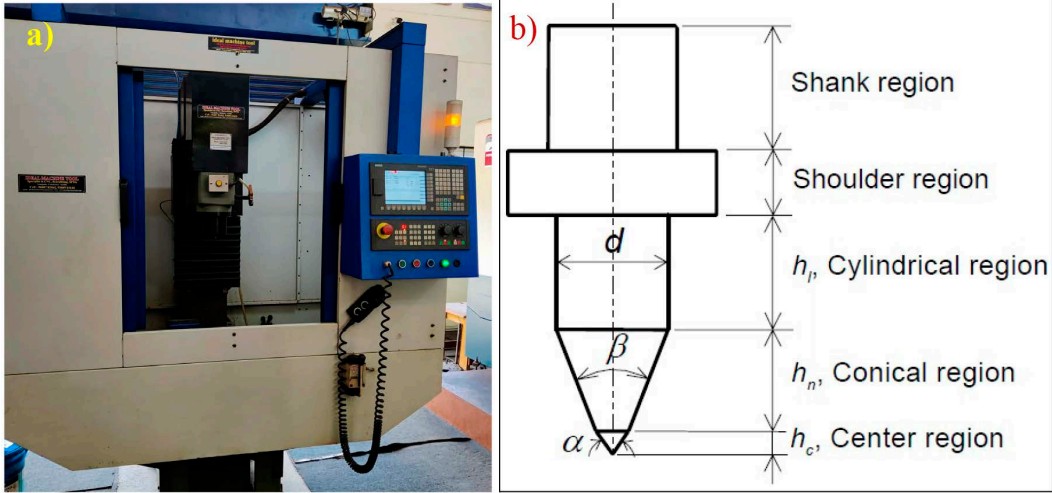

**Fig 2. a)Vertical machining centre, b) Nomenclature of friction drilling tool [5].**

## 3. Results and analysis

Using the Rockwell hardness measuring instrument, the hardness of H13 steel was evaluated. Fig 3 represents the wear loss and hardness for the samples. The untreated sample hardness was very low, resulting in low wear resistance, earlier plastic deformation and abrasive wear. The microstructure is coarse. When the surface is heat treated, the hardness improves due to the changes in microstructure like phase transformation and grain refinement. This helps in severe wear, such as abrasive and adhesion, to mild oxidative wear. The hardness was improved for the AlCrN-coated sample based on heat treatment and ceramic coating. The surface looks micro polish, which helps minimise surface damage and enhance durability. The TiAlN coating exhibited higher hardness and less material loss due to the addition of heat treatment and Titanium Aluminum Nitride. In conclusion, the TiAlN and AlCrN coating has been observed to be a protective barrier to reducing material loss and also helps improve the material's lifespan.

Fig 4 shows the wear performance of H13 steel at varied times. The untreated steel has presented the most wear. The wear rates have shown deviations concerning the elapsed time taken for the run. The wear rate has also been maintained as the temperature has maintained up to 150°C. However, the coated samples have shown a lower wear rate when compared with uncoated and heated samples.

Fig 5 shows the frictional co-efficient results obtained from the samples after 10 minutes. The white line indicates the CoF and the red line indicates the displacement of the probe in a horizontal direction during testing. The CoF plays a vital role in reducing wear rate; lower CoF values have achieved better performance in wear and sliding. The frictional coefficient is observed to be low in coated and increases in uncoated samples. Because of the sample's lubricating effect, a low CoF value is found in coated samples. All samples initially had the same friction coefficient, but the untreated sample's friction coefficient rose due to significant wear.

The abrasive wear is observed in the samples, leading to extensive friction coefficient; the untreated and heated sample CoF values observed are 0.713 and 0.591, and for TiAlN and AlCrN, the CoF values observed are 0.481 and 0.416. The tribological behaviour of the different surface conditions was reflected in the observed values of the Coefficient of Friction (CoF). The untreated sample indicated a strong adhesion and sliding resistance because of a higher CoF value of 0.713. The heated sample observed a slightly reduced CoF of 0.591 by altering the surface with oxidation. The TiAlN coating is unique for hardness, reduced friction through smooth sliding, and low adhesion. A low friction value is observed when TiAlN coating is deposited on polished substrates, which enhances wear resistance. Hence, the surface

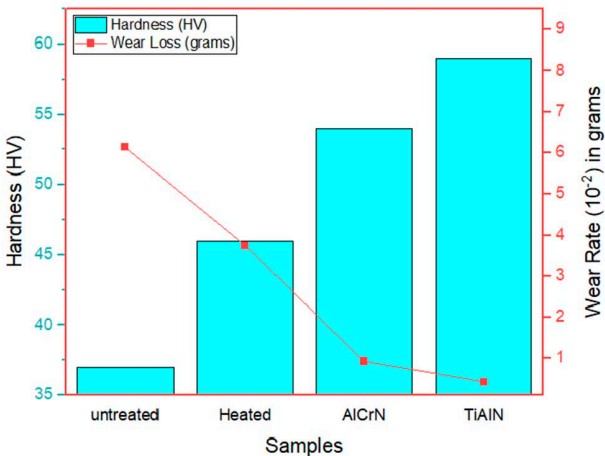

**Fig 3. Wear loss vs. hardness of sample.**

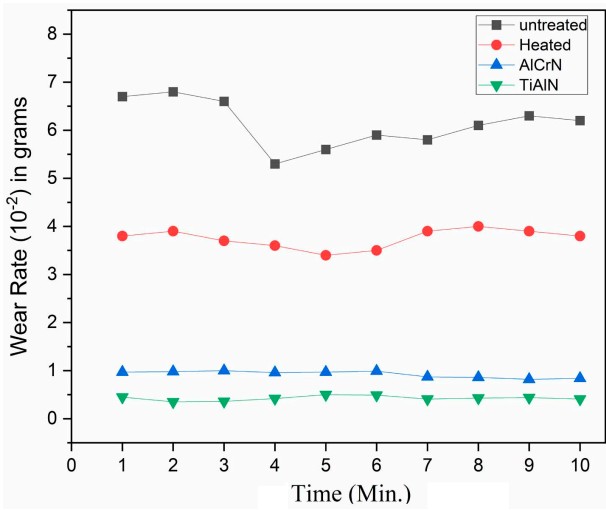

**Fig 4. Wear loss on H13 steel tool.**

performance has resulted in 0.481 of CoF for a TiAlN-coated H13 steel and also because of the lubricious aluminium oxide (Al2O3) trilayer formed at elevated temperature. The AlCrN, with a CoF of 0.416, improves the performance. This helps offer better oxidation resistance, and balanced CoF helps to withstand high load and high-temperature applications.

### 3.1 Surface morphology of wear sample

Scanning electron microscopy images of the wear quality of coated and heat-treated samples are expressed in Fig 6. Micro cracks are visible in all the samples; microgrooves are also visible in a few samples. The depth of the wear is not the same in all samples. However, in Fig 6d, uniform wear is seen, and a few scratches are visible in Fig 6c. The occurrence of plastic deformation and scratches is evident in the samples. Abrasive wear resulted from higher hardness, as seen in Fig 6b. The plastic deformation, oxidation, and material removal resulted from the surface mating at the point of contact; here, due to the coating, the layers are sheltered even when there is a rise in temperature up to 250°C. Due to the plastic deformation, the oxide layers are formed at different places. Traces of adhesive wear were found on the exterior of the uncoated sample and heated sample; these damages are based on the shear and the contact pressure between the metal surfaces. In addition to the findings, a minimal amount of oxide patches can be observed in Fig 6a and 6b, while the hardness difference cannot find any patches in the coated samples. The coated samples covered the oxide layers based on the surface coating and its hardness, even though the temperature was maintained up to 250°C. The surfaces of 6c and 6d were smooth; a glaze-worn surface was observed on the coated sample. The oxide layer, during sliding, may crack and spill off due to its brittle nature, and further sliding may lead to delamination.

The temperature, coating composition and load significantly influence the wear behaviour. The oxidation tendency of coating is determined by temperature. The AlCrN coating forms stable with the help of elevated temperature, which reduces adhesive wear. The TiAlN coating exhibited thermal stability and observed dense microstructure, resulting in oxidation resistance up to 800°C. Meanwhile, the load influences contact pressure and promotes abrasive wear. The load enhances surface adhesion, leading to severe adhesive wear. The coating composition impacts chemical stability, toughness and hardness. The AlCrN coating helps resist the abrasive wear because of its higher hardness, whereas the TiAlN coating, because of its softer surface, offers oxidation resistance and also balances the abrasive and adhesive wear.

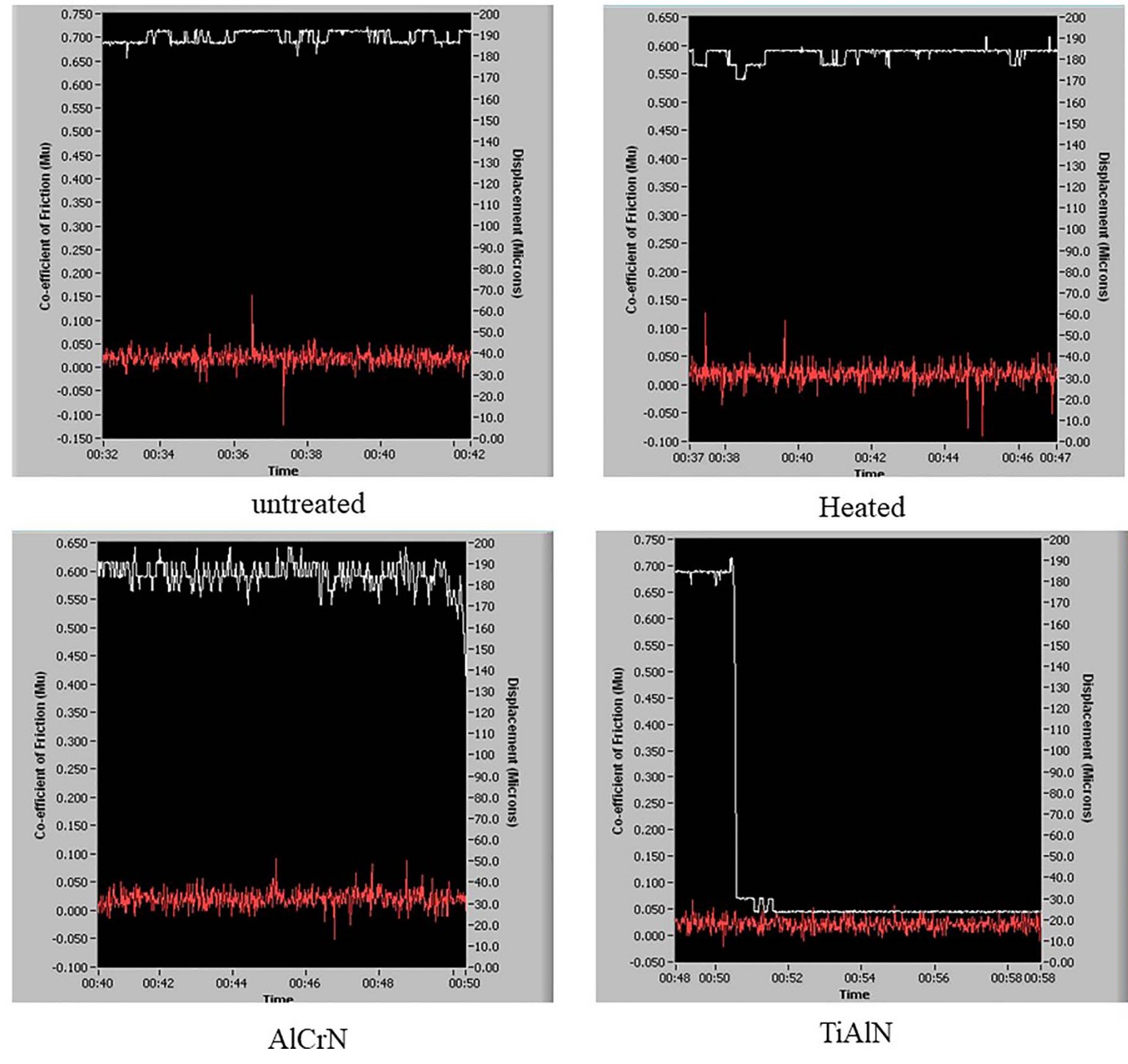

**Fig 5. CoF images of all samples.**

## 3.2 Energy dispersive X-ray (EDS)

Energy Dispersive Spectroscopy (EDS) is used in this work to observe the elemental composition of the samples following the wear study and provide thorough quantitative data. Fig 7 and Table 2 compares the composition of the wear surface and the coated region. As shown in the Fig, a few surface micro-cracks were observed at the initial

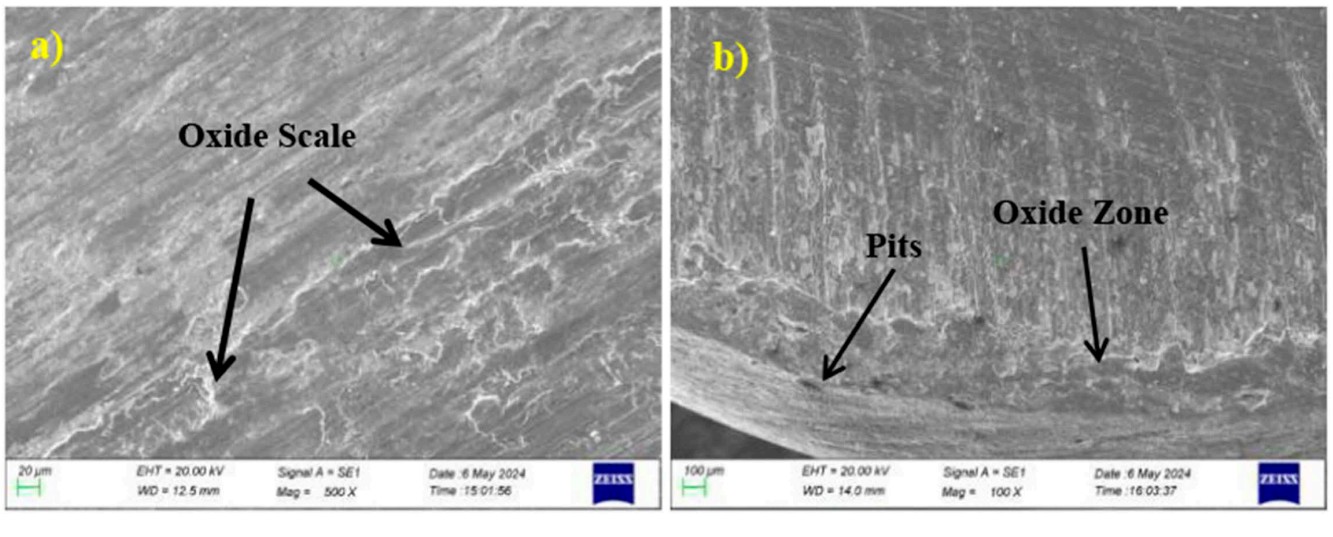

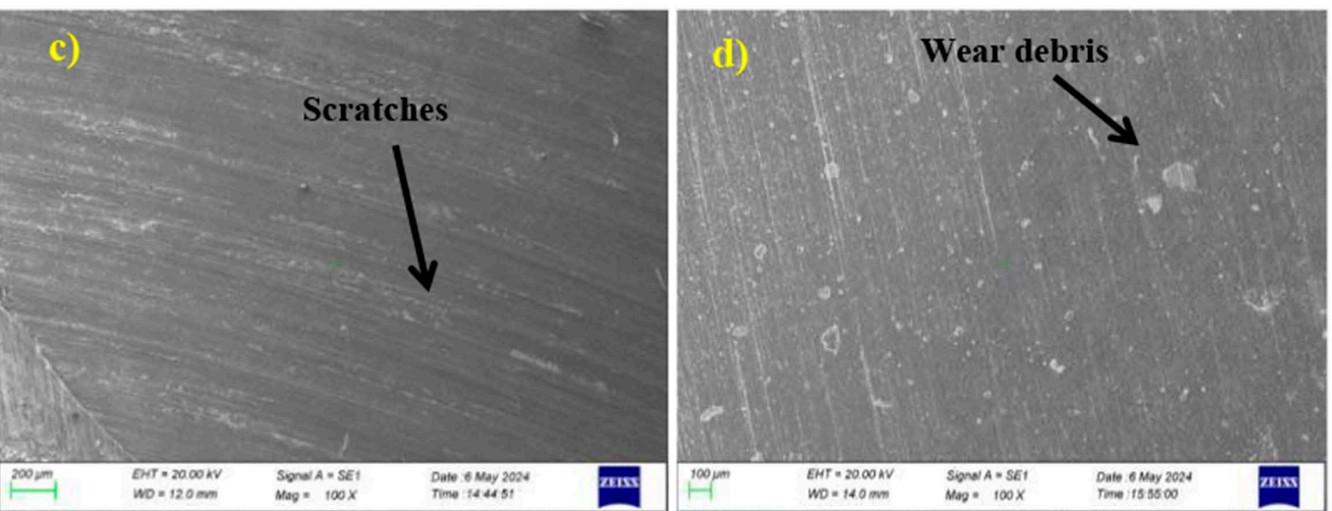

**Fig 6. Surface morphology of all samples a) untreated b) Heated c) AlCrN d) TiAlN.**

stages of the test. Moreover, the spallation of the coating was also visible on the edges of the workplace. Compared with untreated samples, the coated samples have better adhesion properties and wear resistance. The untreated H13 sample's surface exhibits the noticeable adhesive wear depicted in Fig 7a. This results from the untreated steel's decreased hardness compared with the other samples. The EDS inspections have confirmed the presence of iron nitrides based on the existence of nitrogen and iron on the surface. As seen in Fig 7a and 7b, the untreated and hardened samples were discovered to have a high ferrous content; however, the samples' resistance was aided by chromium and molybdenum. The EDS analysis confirmed the nitride layer's existence in sample 7 c- f. This helps improve the hardness of the sub-surface of the coated H13 steel tool. In the coated samples, the existence of coating is observed after the run. However, when comparing the wear and un-wear places for TiAlN and AlCrN coatings, the TiAlN coating demonstrated superior wear resistance, as seen by the absence of H13 components in the worn areas.

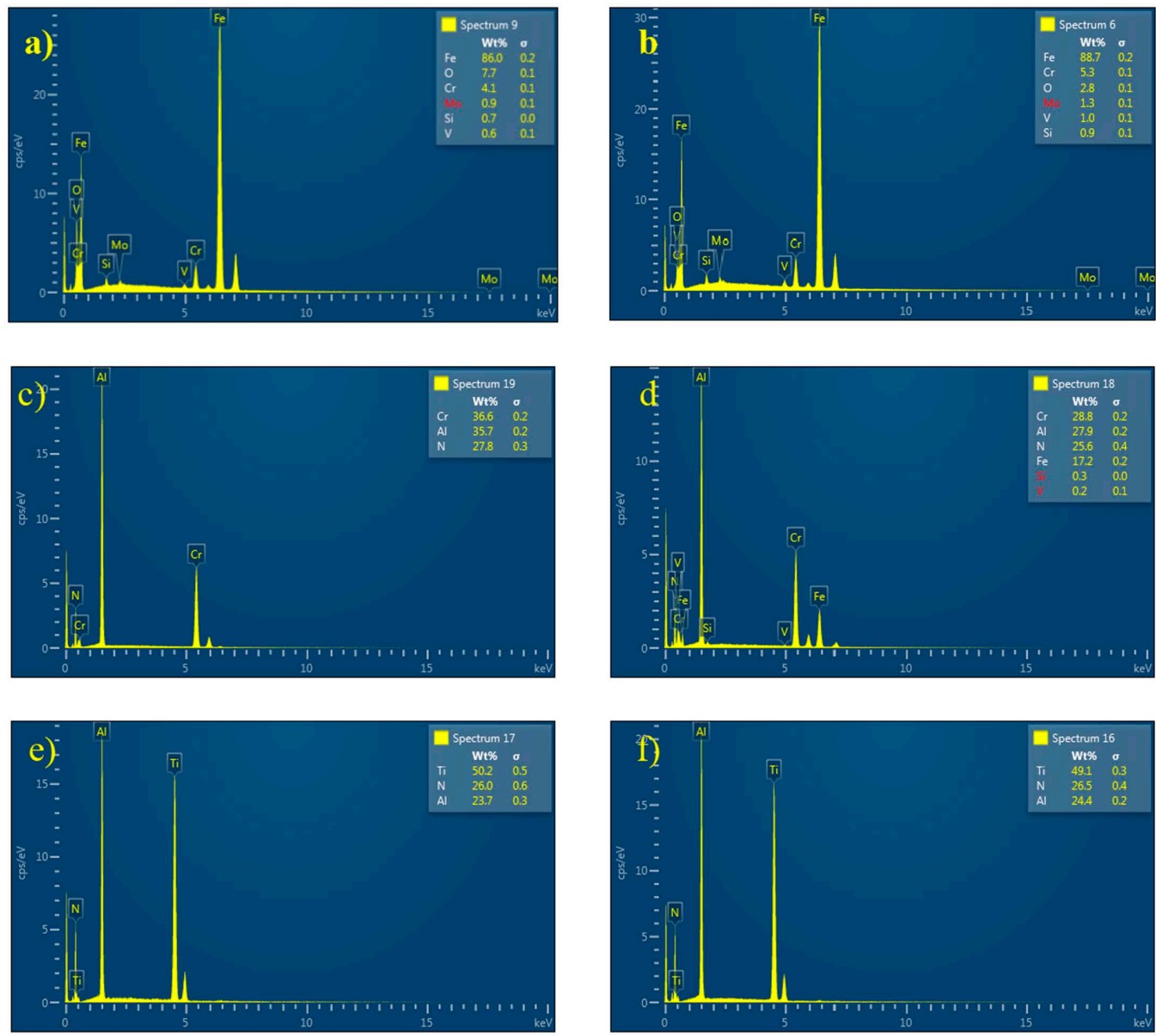

**Fig 7. Composition and morphology of the wear surface of H13 samples a) untreated, b) Hardened, c) AlCrN before wear, d) AlCrN after wear, e) TiAlN before wear f) TiAlN after wear.**

## 3.3 Surface topography

Fig 8 displays the 3D pictures obtained from the Atomic Force Microscope (AFM). After the wear process, the images are captured to study the roughness of the worn surface. The roughness of the surface was captured over the entire work area. The height values along the z-axis were presented statistically. The parameters are primarily defined as Sy (maximum height of Valleys), Sp (maximum height of peaks), Sz (surface height in max.), Sa (Surface arithmetical mean height), and Sq (root mean square of surface).

**Table 2. EDS elemental analysis results.**

| Samples | | Composition Wt % | | | | | | | | |
|---|---|---|---|---|---|---|---|---|---|---|
| | | Al | Ti | N | Fe | O | Cr | Mo | Si | V |
| Untreated | | – | – | – | 86 | 7.7 | 4.1 | 0.9 | 0.7 | 0.6 |
| Heated | | – | – | – | 88.7 | 2.8 | 5.3 | 1.3 | 0.9 | 1.0 |
| Before Wear | AlCrN | 27.9 | – | 25.6 | – | – | 36.6 | – | – | – |
| | TiAlN | 23.7 | 50.2 | 26 | – | – | – | – | – | – |
| After Wear | AlCrN | 27.9 | – | 25.6 | 17.2 | – | 28.8 | – | 0.3 | 0.2 |
| | TiAlN wear | 24.4 | 49.1 | 26.5 | – | – | – | – | – | – |

Atomic Force Microscopy (AFM) is used to observe the sample's surface roughness; Figs 6(b) and 8(a) illustrate the roughness values. It is understood that the coatings exhibited roughness compared to the face of the samples. The roughness is high in the first two samples, mainly on the untreated sample; better wear resistance was observed because of improving hardness.

The surface values are shown in Table 3. The AlCrN and TiAlN coating has exhibited less exterior roughness than untreated and heated steel, as shown in Fig 8(c) and 8(d). Both the coating surfaces contain higher levels of deposition TiAlN and AlCrN using the PVD technique, resulting in elevated roughness. During cathodic deposition, the material evaporates at a lower melting point, which leads to more particles with larger size and volume. The untreated sample presented significantly less hardness but was relatively brittle. For the entry of oxygen, cracks were found near the damaged surface, which may guide the development of oxides finally into delamination. The plastic behaviour leads to less surface deformation; the heated sample has high hardness, which is less brittle. When compared to the untreated sample, there are fewer wear cracks. The coated sample has strength, toughness, and very fracture resistance, which will lead to improvement.

### 3.4 Surface morphology of friction drilling tool

The friction drilling process was carried out at a spindle speed of 3000 rpm and a feed rate of 0.1 mm/rev. The AlCrN-coated tool has a better surface finish than the TiAlN-coated tool [41]. Even after continuous running, the nanocrystalline layer is visible on the surface of both tools. The bonding between the tool and the surface is deviated in a few places. Fig 9 shows the SEM morphologies of the Coated H13 steel tool. The worn surface of the TiAlN-coated H13 steel tool is visible in Fig 9(a). The image represents the typical oxidation wear characteristics and delamination in the centre areas where the temperature was higher in that region. The oxidation wear can be visible only on the grey side walls. Peeling is visible in the central region of the TiAl-coated tool material. However, the material peeling is very low, as shown in Fig 9(b). the bonding between the tool and the coating is evident in the TiAlN-coated friction drilling tool.

### 3.5 Surface morphology of AZ31B

The surface morphology of Friction drilling AZ31B material of TiAlN and AlCrN and a coated tool is shown in Fig 10(a) and 10(b). The chips melt, and the additionals help to form a bushing in the surface. In the image, there is no evidence of edge serrations, and a cylindrical portion of the tool helps to finish the surface with very little roughness. Because of the heat generation and high friction, a strong adhesion is observed on the AZ31B magnesium alloy in the deformation zone. The crack growth is prevented because of the ductile nature of the AZ31B. The plastic deformation is observed due to the material's heat behaviour, which also helps promote early yielding.

### 3.6 Friction drilling analysis

The friction drilled tool and bushing formed, and the conical shape is shown in Fig 11(a–d). The formation of the bushing and the surface roughness [37] became evident in the above Fig. The roughness exhibited a smoother surface finish at

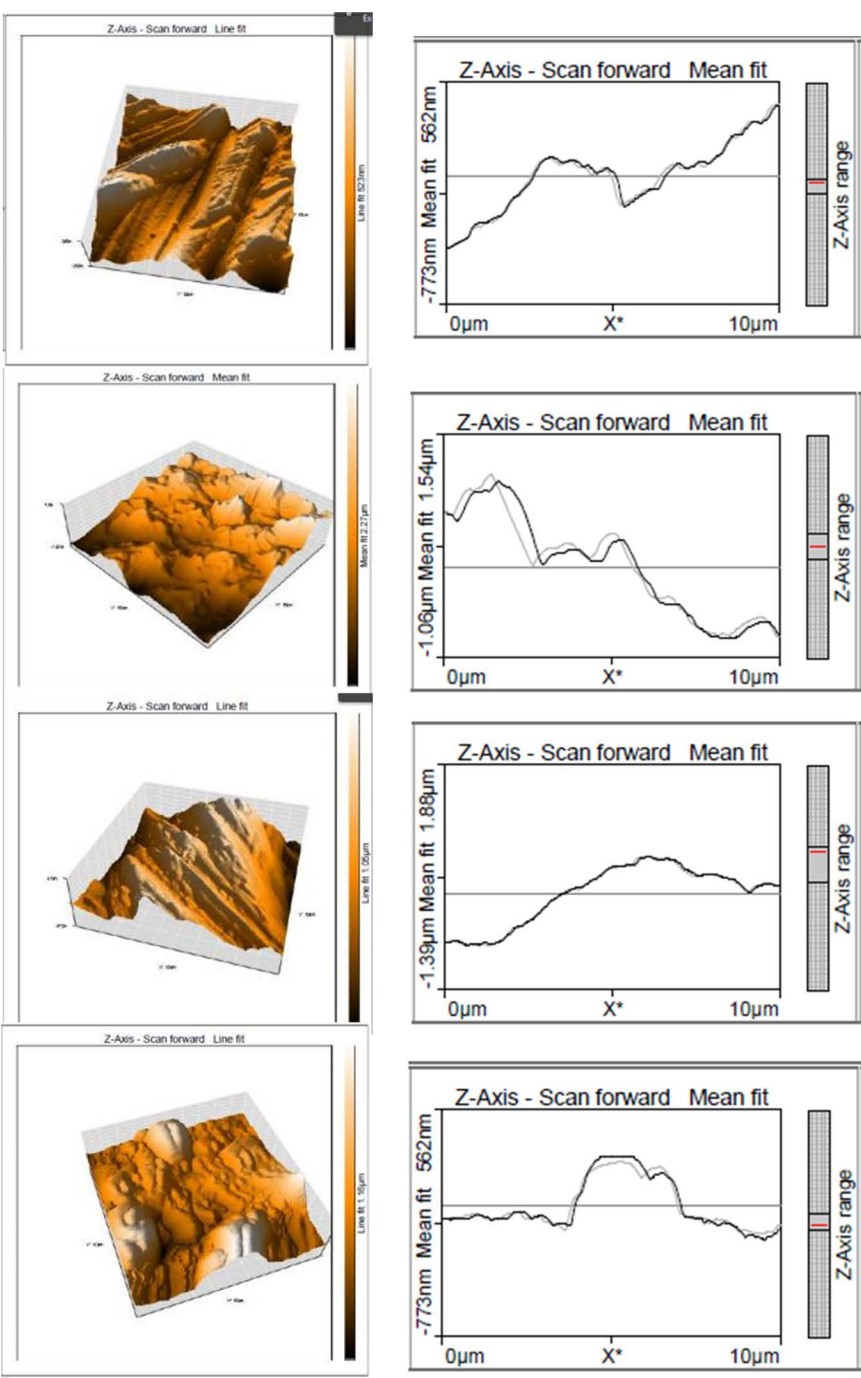

**Fig 8. The 3D profile optical profilometer.**

3000 rpm and 0.1 mm/rev. The bushing height is also well formed during the process because of the heat generated and material flow. This process involves significant thermal and mechanical interactions based on the selection of a lower feed rate selection. The high spindle speed and the lower feed rate help form the material's bushing quality.

**Table 3. Surface values using Atomic Force Microscopy (AFM).**

| Parameters | Untreated | Heated | AlCrN | TiAlN |
|---|---|---|---|---|
| $S_a$ (nm) | 72.631 | 169.26 | 164.869 | 148.75 |
| $S_q$ (nm) | 87.404 | 211.25 | 199.03 | 193.13 |
| $S_y$ (nm) | 532.48 | 1687.8 | 1053.8 | 1184.2 |
| $S_p$ (nm) | 262.43 | 792.19 | 479.26 | 574.2 |

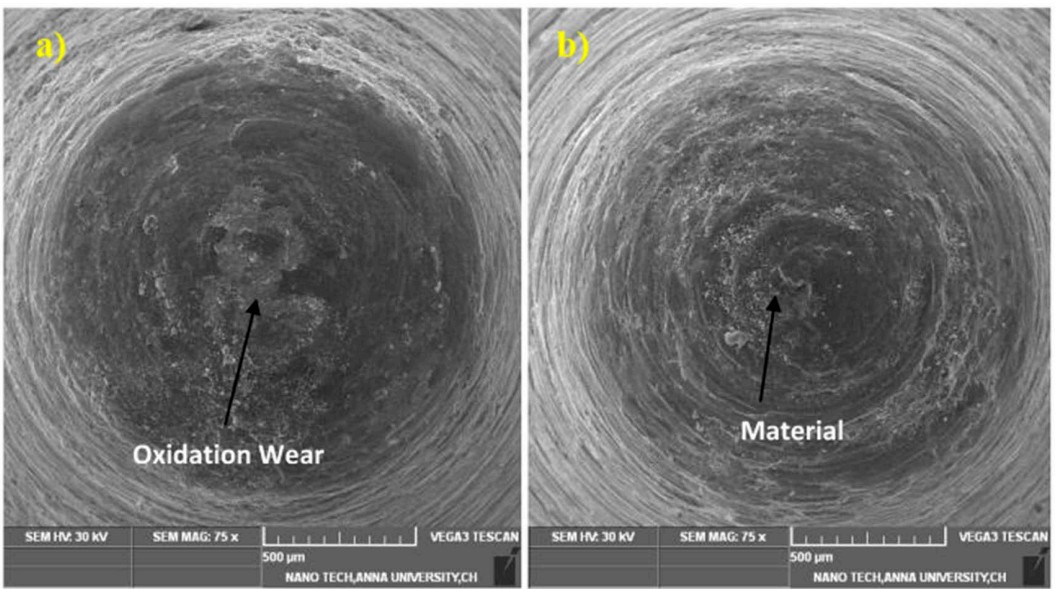

**Fig 9. Friction drilling tool a) TiAlN coated tool, b) AlCrN coated tool.**

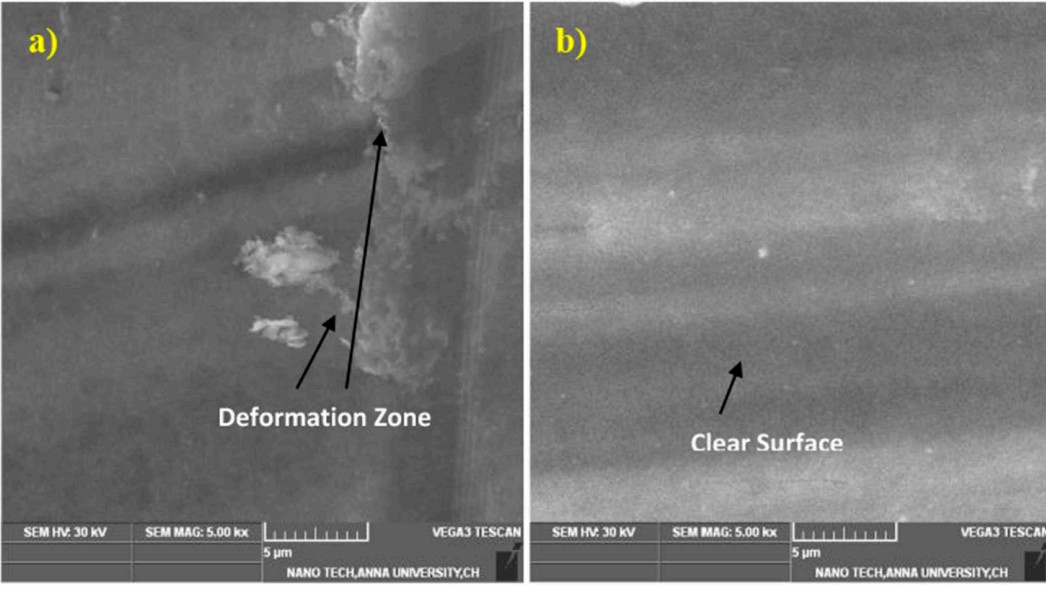

**Fig 10. Surface morphology of AZ31B material a) TiAlN coated tool, b) AlCrN coated tool.**

**Fig 11. a) Friction drilling tool, b) Bushing formation, c&d) Cylindricity.**

## 4. Conclusion

The experimental study's results were used to make the findings. The wear resistance and friction between the samples are analyzed under a variety of settings. Wear resistance strongly depends on features such as adhesive wear and oxidation wear. The steel's hardness influences the choice of surface treatment. The coating aids in lowering delamination and fracture resistance.

• Adhesive wear and oxidation wear were noted for the heated and untreated samples.

• TiAlN and AlCrN coatings show lower wear rates when compared to other samples. The sample coated with TiAlN showed less wear than the one covered with AlCrN.

• For TiAlN and AlCr-coated H13, the material loss was reported to be 0.00013 and 0.00079 grams, respectively.

• The samples show extensive abrasive wear, leading to a high friction coefficient; the untreated and heated sample CoF values are 0.713 and 0.591, and for TiAlN and AlCrN, the CoF values are 0.481 and 0.416.

- The bushing height was also well formed at 3000 rpm and 0.1 mm/rev.

- Due to the ductile nature of the material, very minimal cracks were observed on the surface of AZ31B.

The future study will be focused on

- Compared with other surface treatments for friction drilling, the H13 steel tool is the best.

- To validate the performance of the coated friction tool via testing with industry scales.

## Author contributions

**Conceptualization:** Saravanan Balakrishnan.

**Data curation:** Emad Abouel Nasr.

**Formal analysis:** Saravanan Balakrishnan, Robert Čep, Emad Abouel Nasr.

**Funding acquisition:** Saravanan Balakrishnan, Robert Čep, Sachin Salunkhe.

**Investigation:** Robert Čep, Sachin Salunkhe.

**Methodology:** Saravanan Balakrishnan, Selvakumar Subbaiah, Mathew Alphonse, Sachin Salunkhe.

**Project administration:** Robert Čep, Sachin Salunkhe.

**Resources:** Mathew Alphonse.

**Software:** Selvakumar Subbaiah, Robert Čep, Emad Abouel Nasr.

**Supervision:** Selvakumar Subbaiah, Mathew Alphonse, Robert Čep, Emad Abouel Nasr.

**Validation:** Saravanan Balakrishnan, Selvakumar Subbaiah, Mathew Alphonse, Robert Čep, Emad Abouel Nasr.

**Visualization:** Saravanan Balakrishnan, Sachin Salunkhe, Emad Abouel Nasr.

**Writing – original draft:** Saravanan Balakrishnan, Selvakumar Subbaiah, Mathew Alphonse, Sachin Salunkhe, Emad Abouel Nasr.

**Writing – review & editing:** Mathew Alphonse.

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
