## [Decision Letter · Decision Letter 0]

2 May 2025

Dear Dr. Salunkhe,

Thank you for submitting your manuscript to PLOS ONE. After careful consideration, we feel that it has merit but does not fully meet PLOS ONE’s publication criteria as it currently stands. Therefore, we invite you to submit a revised version of the manuscript that addresses the points raised during the review process.

**ACADEMIC EDITOR:**

As per reviewers' feedback the authors need to revise their manuscript and take the task of major revision.

We look forward to receiving your revised manuscript.

Kind regards,

Himadri Majumder, Ph.D

Academic Editor

PLOS ONE

“The authors also extend their appreciation to King Saud University for funding the publication of this work through Researchers Supporting Project number (RSP2025R164), King Saud University, Riyadh, Saudi Arabia. This article was co-funded by the European Union under the REFRESH – Research Excellence For Region Sustainability and High-tech Industries project number CZ.10.03.01/00/22_003/0000048 via the Operational Programme Just Transition and has been done in connection with project Students Grant Competition SP2024/087 “Specific Research of Sustainable Manufacturing Technologies” financed by the Ministry of Education, Youth and Sports and Faculty of Mechanical Engineering VŠB-TUO. Article has been done in connection with project Students Grant Competition SP2024/087 “Specific Research of Sustainable Manufacturing Technologies” financed by the Ministry of Education, Youth and Sports and Faculty of Mechanical Engineering VŠB-TUO.”

“The authors also extend their appreciation to King Saud University for funding the publication of this work through Researchers Supporting Project number (RSP2025R164), King Saud University, Riyadh, Saudi Arabia. This article was co-funded by the European Union under the REFRESH – Research Excellence For Region Sustainability and High-tech Industries project number CZ.10.03.01/00/22_003/0000048 via the Operational Programme Just Transition and has been done in connection with project Students Grant Competition SP2024/087 “Specific Research of Sustainable Manufacturing Technologies” financed by the Ministry of Education, Youth and Sports and Faculty of Mechanical Engineering VŠB-TUO. Article has been done in connection with project Students Grant Competition SP2024/087 “Specific Research of Sustainable Manufacturing Technologies” financed by the Ministry of Education, Youth and Sports and Faculty of Mechanical Engineering VŠB-TUO.”

Reviewers' comments:

Reviewer's Responses to Questions

**Comments to the Author**

1. Is the manuscript technically sound, and do the data support the conclusions?

Reviewer #1: Yes

Reviewer #2: No

Reviewer #3: Yes

Reviewer #4: Yes

2. Has the statistical analysis been performed appropriately and rigorously?

Reviewer #1: Yes

Reviewer #2: N/A

Reviewer #3: Yes

Reviewer #4: Yes

3. Have the authors made all data underlying the findings in their manuscript fully available?

Reviewer #1: Yes

Reviewer #2: No

Reviewer #3: Yes

Reviewer #4: Yes

4. Is the manuscript presented in an intelligible fashion and written in standard English?

Reviewer #1: Yes

Reviewer #2: No

Reviewer #3: Yes

Reviewer #4: Yes

Reviewer #1: Comments for the Author:

Authors have focused on interesting area. The paper is well written and structured.

Comment 1:

The coefficient of friction (COF) for the AlCrN coating is reported as 0.046; however, in Figure 3, the observed value appears to be approximately 0.7. The authors are requested to verify these findings and make the necessary corrections.

Comment 2:

Compared to untreated and heat-treated specimens, the AlCrN-coated samples show a decrease in wear resistance due to increased hardness. While this explanation is logically acceptable, it raises a question: how is a reduction in the coefficient of friction observed for these coated samples? A scientific explanation for this phenomenon is required. Although improved hardness is generally associated with better wear resistance, it does not directly correlate with a reduction in the coefficient of friction.

Comment 3:

In Figure 2, wear loss is plotted against time for different samples. However, the wear loss appears to remain constant over time. Typically, wear loss is expected to increase as time progresses. The authors are requested to clarify this observation. Please also check and revise the English and grammar used in the figure caption and related discussion.

Comment 4:

The references cited in the manuscript appear to be relevant and up-to-date. However, it might be beneficial to ensure that a broader range of studies, can be added such as

https://doi.org/10.1080/10426914.2024.2323437

https://doi.org/10.1177/0267084424124112

https://doi.org/10.1016/j.matpr.2022.12.062.

DOI: 10.18185/erzifbed.430628

https://doi.org/10.1115/1.4051225

https://doi.org/10.1016/j.ceramint.2020.04.015

https://doi.org/10.1007/s40430-020-02721-8.

Comment 5:

The manuscript needs to check for English language and grammatical correction.

Reviewer #2: 1. In abstract, “where carried out” should be “were carried out”; “Oxditation” should be “Oxidation”. In the introduction, “subtract” instead of “substrate”; “steelies” instead of “steel’s”. Similarly, the manuscript contains numerous grammatical errors and typos.

2. The friction coefficient (COF) values, like 0.014 for TiAlN, are suspiciously low and potentially incorrect.

3. Coefficient of friction values reported in the abstract, results, and conclusion vary. For example, 0.417 in one place, 0.147 in another).

4. The sliding distance mentioned in the experimental procedure is 1000 m. Is it correct?

5. The PVD method is mentioned but not explained well. Details about deposition parameters (duration, pressure, thickness) are missing. So, add a brief explanation of the coating technique (PVD – sputtering/evaporation) and specify process conditions (deposition time, substrate temperature, etc.).

6. Repetitive figure labels. For example, multiple "Figure 2".

7. In the results and analysis section, graphs and images are described without deep analysis. So, improve data interpretation by relating wear patterns to hardness and microstructural changes.

8. Clearly explain how temperature, load, and coating composition relate to observed wear types (abrasive, adhesive, and oxidation).

9. The element content isn't quantified (e.g., atomic% % of Fe, N). So, provide a table summarising the EDS elemental analysis results.

10. The image quality of most of the figures is very poor. Also, the labelling is not clear. For example, in Fig. 5, sublabels are not very clear, and in Fig. 6, sublabels are not available. So, check all the figures properly and provide a better quality image along with proper labelling.

11. In the result and analysis section, no data is reported with interpreting trends or connecting with previous literature.

12. There are numerous technical errors throughout the manuscript. For instance, in Table 2, the parameter "Sy" is listed but not defined or discussed in the text—there is no explanation of what "Sy" represents, nor how it relates to surface roughness or wear behaviour.

Reviewer #3: The article is very interesing and industry relevant.

1. The research novelty should be clearer indicated.

2. Figure 1: The photo has low quality; improve it

3. The introduction should emphasize the novelty of the given approach.

4. Figure 2: The photo has low quality; improve it

5. Figure 3: Legend is missing

6. Figure 7: Legend is missing

7. Figure 8: Legend is missing

8. The abstract should include information about new methods, results, concepts, and conclusions. Author need to rewrite the abstract in its current form to incorporate more information about the achievements described in the manuscript.

9. In tribological analysis, the material pair should be given in the drawings. Why was the disk material different from the processed material?

10. The roughness parameter should be specified

11. The study should discuss the practical impact of coating on drilling tool performance. How does the improved wear resistance translate into tool life enhancement? More discussion on industrial applications would strengthen the manuscript.

12. The conclusion summarizes key findings but lacks a clear direction for future research.

Reviewer #4: 1. The tool photograph is shown, but the tool design is not shown or discussed. Provide the tool design or its citation.

2. Check the English language presentation.

3. Include note on bosh and bush formation.

4. Highlight the need for using H13 as tool material

**Do you want your identity to be public for this peer review?** For information about this choice, including consent withdrawal, please see our Privacy Policy

Reviewer #1: **Yes: ** Avinash Borgaonkar

Reviewer #2: No

Reviewer #3: **Yes: ** Amlana Panda

Reviewer #4: **Yes: ** V.K.BUPESH RAJA

---

## [Author Response · Author response to Decision Letter 1]

20 May 2025

Title: Investigation of Wear Behaviour and Surface Analysis of a Coated H13 Material for Friction Drilling Application

Reviewer Comments & Authors Response

Reviewer 1

Sl. No. Reviewer Comments Authors Response

1 The coefficient of friction (COF) for the AlCrN coating is reported as 0.046; however, in Figure 3, the observed value appears to be approximately 0.7. The authors are requested to verify these findings and make the necessary corrections. Thanks for pointing out the mistake. The Coefficient of Friction (CoF) values are corrected.

“Recorded on Page No. 2, 9 & 19.”

2 Compared to untreated and heat-treated specimens, the AlCrN-coated samples show a decrease in wear resistance due to increased hardness. While this explanation is logically acceptable, it raises a question: how is a reduction in the coefficient of friction observed for these coated samples? A scientific explanation for this phenomenon is required. Although improved hardness is generally associated with better wear resistance, it does not directly correlate with a reduction in the coefficient of friction. The tribological behaviour of the different surface conditions were reflected in the observed values of Coefficient of Friction (CoF). The untreated sample indicating a strong adhesion as well as sliding resistance because of higher CoF value of 0.713. The heated sample observed little reduced CoF of 0.591, by altering the surface with the presence of oxidation. The TiAlN coating is unique for high hardness; reduce friction through smooth sliding and low adhesion. A low friction value is observed when TiAlN coating is deposited on polished substrates which enhance wear resistance. Hence the surface performance has resulted in 0.046 of CoF for a TiAlN coated H13 steel and also because of a lubricious aluminium oxide (Al₂O₃) tribolayer formed at elevated temperature. The AlCrN, with a CoF of 0.416 improves the performance. This helps in offering better oxidation resistance and balanced CoF helps in withstanding high load as well as high temperature applications.

3 In Figure 2, wear loss is plotted against time for different samples. However, the wear loss appears to remain constant over time. Typically, wear loss is expected to increase as time progresses. The authors are requested to clarify this observation. Please also check and revise the English and grammar used in the figure caption and related discussion. Thanks for pointing out the mistake, Figure 4 updated

“Recorded on Page No. 8.”

4 The references cited in the manuscript appear to be relevant and up-to-date. However, it might be beneficial to ensure that a broader range of studies, can be added such as

https://doi.org/10.1080/10426914.2024.2323437

https://doi.org/10.1177/0267084424124112

https://doi.org/10.1016/j.matpr.2022.12.062.

DOI: 10.18185/erzifbed.430628

https://doi.org/10.1115/1.4051225

https://doi.org/10.1016/j.ceramint.2020.04.015

https://doi.org/10.1007/s40430-020-02721-8.

All the articles cited except this because of an error (https://doi.org/10.1177/0267084424124112)

5 The manuscript needs to check for English language and grammatical correction. Thanks for pointing out. The entire manuscript has carefully revised.

Reviewer 2

Sl. No. Reviewer Comments Authors Response

1 In abstract, “where carried out” should be “were carried out”; “Oxditation” should be “Oxidation”. In the introduction, “subtract” instead of “substrate”; “steelies” instead of “steel’s”. Similarly, the manuscript contains numerous grammatical errors and typos. Thanks for pointing out. The entire manuscript has carefully revised.

2 The friction coefficient (COF) values, like 0.014 for TiAlN, are suspiciously low and potentially incorrect. The TiAlN coating is unique for high hardness; reduce friction through smooth sliding and low adhesion. A low friction value is observed when TiAlN coating is deposited on polished substrates which enhance wear resistance. Hence the surface performance has resulted in 0.046 of CoF for a TiAlN coated H13 steel.

3 Coefficient of friction values reported in the abstract, results, and conclusion varies. For example, 0.417 in one place, 0.147 in another). Thanks for pointing out. The entire manuscript has carefully revised.

“Recorded on Page No. 2, 9 & 19.”

4 The sliding distance mentioned in the experimental procedure is 1000 m. Is it correct? Thanks for pointing out. The correct sliding distance is 200 m

“Recorded on Page No. 5.”

5 The PVD method is mentioned but not explained well. Details about deposition parameters (duration, pressure, thickness) are missing. So, add a brief explanation of the coating technique (PVD – sputtering/evaporation) and specify process conditions (deposition time, substrate temperature, etc.).

Thanks for pointing out. The experimental procedure added.

“Recorded on Page No. 5.”

6 Repetitive figure labels. For example, multiple "Figure 2". Thanks for pointing out the mistake. Now its corrected

7 In the results and analysis section, graphs and images are described without deep analysis. So, improve data interpretation by relating wear patterns to hardness and microstructural changes. Thanks for pointing out the mistake. Now its corrected

8 Clearly explain how temperature, load, and coating composition relate to observed wear types (abrasive, adhesive, and oxidation).

The wear behaviour significantly influenced by the temperature, coating composition and load. The oxidation tendency of coating is determined by temperature. The AlCrN coating forms stable with the help of elevated temperature which helps in reducing the adhesive wear. The TiAlN coating exhibited thermal stability also can observed dense microstructure which result in oxidation resistance upto 800 °C. Whereas the load influences the contact pressure, promotes abrasive wear. The load enhances surface adhesion leads to severe adhesive wear. The coating composition impacts chemical stability, toughness and hardness. The AlCrN coating helps in resist the abrasive wear, because of higher hardness, whereas the TiAlN coating because of softer surface offers oxidation resistance also balance the abrasive and adhesive wear

9 The element content isn't quantified (e.g., atomic% % of Fe, N). So, provide a table summarising the EDS elemental analysis results. Thanks for pointing out

“Recorded on Page No. 13.”

10 The image quality of most of the figures is very poor. Also, the labelling is not clear. For example, in Fig. 5, sublabels are not very clear, and in Fig. 6, sublabels are not available. So, check all the figures properly and provide a better quality image along with proper labelling. Thanks for pointing out

Figure quality improved

11 In the result and analysis section, no data is reported with interpreting trends or connecting with previous literature. Added

12 There are numerous technical errors throughout the manuscript. For instance, in Table 2, the parameter "Sy" is listed but not defined or discussed in the text—there is no explanation of what "Sy" represents, nor how it relates to surface roughness or wear behaviour. Now it’s done

“Recorded on Page No. 14.”

Reviewer 3

Sl. No. Reviewer Comments Authors Response

1 The research novelty should be clearer indicated. Thanks for pointing out, Novelty added in the Abstract as well as introduction

“Recorded on Page No. 2”

2 Figure 1: The photo has low quality; improve it Improved

“Recorded on Page No.6”

3 The introduction should emphasize the novelty of the given approach. Novelty added in the Abstract as well as introduction

4 Figure 2: The photo has low quality; improve it Improved

“Recorded on Page No.6”

5 Figure 3: Legend is missing Thanks for pointing out, now it’s added

6 Figure 7: Legend is missing Thanks for pointing out, now it’s added

7 Figure 8: Legend is missing Thanks for pointing out, now it’s added

8 The abstract should include information about new methods, results, concepts, and conclusions. Author need to rewrite the abstract in its current form to incorporate more information about the achievements described in the manuscript. Thanks for pointing out, now it’s added

9 In tribological analysis, the material pair should be given in the drawings. Why was the disk material different from the processed material? The Disc used for this study is EN 31 with higher hardness of 62 HRC.

10 The roughness parameter should be specified Added in the session 3.3

“Recorded on Page No.14”

11 The study should discuss the practical impact of coating on drilling tool performance. How does the improved wear resistance translate into tool life enhancement? More discussion on industrial applications would strengthen the manuscript. Thanks for pointing out, the practical impact of nitriding and its industrial applications have already been discussed in the introduction. To strengthen the discussion, a few additional points were also included

12 The conclusion summarizes key findings but lacks a clear direction for future research Thanks for pointing out, now it’s added

Reviewer 4

Sl. No. Reviewer Comments Authors Response

1 The tool photograph is shown, but the tool design is not shown or discussed. Provide the tool design or its citation. Thanks for pointing out, added in Figure 2 b

“Recorded on Page No.7”

2 Check the English language presentation. The entire manuscript checked

3 Include note on bosh and bush formation. Thanks for pointing out, now it’s added

“Recorded on Page No.4”

4 Highlight the need for using H13 as tool material Thanks for pointing out, now it’s added

“Recorded on Page No.4”

---

## [Decision Letter · Decision Letter 1]

12 Jun 2025

Dear Dr. Salunkhe,

Thank you for submitting your manuscript to PLOS ONE. After careful consideration, we feel that it has merit but does not fully meet PLOS ONE’s publication criteria as it currently stands. Therefore, we invite you to submit a revised version of the manuscript that addresses the points raised during the review process.

We look forward to receiving your revised manuscript.

Kind regards,

Himadri Majumder, Ph.D

Academic Editor

PLOS ONE

Journal Requirements:

Reviewers' comments:

Reviewer's Responses to Questions

**Comments to the Author**

Reviewer #1: All comments have been addressed

Reviewer #3: All comments have been addressed

2. Is the manuscript technically sound, and do the data support the conclusions?

Reviewer #1: Yes

Reviewer #3: Yes

3. Has the statistical analysis been performed appropriately and rigorously?

Reviewer #1: Yes

Reviewer #3: Yes

4. Have the authors made all data underlying the findings in their manuscript fully available?

Reviewer #1: Yes

Reviewer #3: Yes

5. Is the manuscript presented in an intelligible fashion and written in standard English?

Reviewer #1: Yes

Reviewer #3: Yes

Reviewer #1: Comment to author

Comment 1: The time considered for measurement of wear rate shown in figure 4 about 10 second is very short. Why is author selected such shorter period to evaluate the wear rate of the specimen?

Comment 2: Same for the coefficient of friction as shown in figure 5 about 10 second is very short. Why is author selected such shorter period?

Comment 3: In Fig 5, The white line indicates the CoF and the red line indicates the displacement of the probe in a horizontal direction during testing. For untreated, heated and AlCrN samples both signals are exhibiting similar kind of trend. But for TiAlN why CoF is lowered drastically, can author explain physics behind this?

Comment 4: Fig 11 b shows the formation of the bushing and the surface roughness, is there any technique which can be implemented to avoid or reduce the formation of the bushing?

Comment 5: While the references cited in the manuscript are relevant and reasonably up-to-date, it would strengthen the work to include a broader range of recent studies in the field to ensure comprehensive coverage of current advancements.

https://doi.org/10.3390/ma16041594

doi:10.1177/13506501231184304.

https://doi.org/10.1007/s42452-019-1152-6.

https://doi.org/10.1007/s12046-020-1266-y.

Reviewer #3: In overall, the organization of the paper is good and worthy of publication. Researchers have answered all the comments.

Now the paper is accepted and recommended for publication.

**Do you want your identity to be public for this peer review?** For information about this choice, including consent withdrawal, please see our Privacy Policy

Reviewer #1: No

Reviewer #3: No

---

## [Author Response · Author response to Decision Letter 2]

14 Jun 2025

Title: Investigation of Wear Behaviour and Surface Analysis of a Coated H13 Material for Friction Drilling Application

Reviewer Comments & Authors Response

Reviewer 1

Sl. No. Reviewer Comments Authors Response

1 The time considered for measurement of wear rate shown in figure 4 about 10 second is very short. Why author is selected such shorter period to evaluate the wear rate of the specimen? Thanks for pointing out the mistake. The wear rate taken is for 10 Minutes only. It is already mentioned in chapter 2 experiment procedure.

2 Same for the coefficient of friction as shown in figure 5 about 10 second is very short. Why author is selected such shorter period? Thanks for pointing out the mistake, the wear study has done for 10 minutes only.

Updated in manuscript

3 In Fig 5, The white line indicates the CoF and the red line indicates the displacement of the probe in a horizontal direction during testing. For untreated, heated and AlCrN samples both signals are exhibiting similar kind of trend. But for TiAlN why CoF is lowered drastically, can author explain physics behind this? Figure updated, Thanks

4 Fig 11 b shows the formation of the bushing and the surface roughness, is there any technique which can be implemented to avoid or reduce the formation of the bushing? In friction drilling, bushing formation can be reduced by optimizing speed, tool geometry, lubrication, and backing support—but bushing remains a key factor for thread engagement and joint strength.

5 While the references cited in the manuscript are relevant and reasonably up-to-date, it would strengthen the work to include a broader range of recent studies in the field to ensure comprehensive coverage of current advancements.

https://doi.org/10.3390/ma16041594

doi:10.1177/13506501231184304.

https://doi.org/10.1007/s42452-019-1152-6.

https://doi.org/10.1007/s12046-020-1266-y.

All the references cited in the manuscript, thanks for the support.

---

## [Decision Letter · Decision Letter 2]

27 Jun 2025

Investigation of Wear Behaviour and Surface Analysis of a Coated H13 Material for Friction Drilling Application

PONE-D-25-17191R2

Dear Dr. Salunkhe,

We’re pleased to inform you that your manuscript has been judged scientifically suitable for publication and will be formally accepted for publication once it meets all outstanding technical requirements.

Kind regards,

Himadri Majumder, Ph.D

Academic Editor

PLOS ONE

Additional Editor Comments (optional):

Reviewers' comments:

Reviewer's Responses to Questions

**Comments to the Author**

Reviewer #1: All comments have been addressed

2. Is the manuscript technically sound, and do the data support the conclusions?

Reviewer #1: Yes

3. Has the statistical analysis been performed appropriately and rigorously?

Reviewer #1: Yes

4. Have the authors made all data underlying the findings in their manuscript fully available?

Reviewer #1: Yes

5. Is the manuscript presented in an intelligible fashion and written in standard English?

Reviewer #1: Yes

Reviewer #1: The authors have carefully addressed all reviewer comments and incorporated the suggestions. The manuscript focuses on an interesting and relevant research area, which has been further strengthened in the revision.

**Do you want your identity to be public for this peer review?** For information about this choice, including consent withdrawal, please see our Privacy Policy

Reviewer #1: No

---

## [Editor Report · Acceptance letter]

PONE-D-25-17191R2

PLOS ONE

Dear Dr. Salunkhe,

I'm pleased to inform you that your manuscript has been deemed suitable for publication in PLOS ONE. Congratulations! Your manuscript is now being handed over to our production team.

Kind regards,

on behalf of

Dr. Himadri Majumder

Academic Editor

PLOS ONE